# Using ISARIC 4C mortality score to predict dynamic changes in mortality risk in COVID-19 patients during hospital admission

Tim Crocker-Buque[1]⊕*, Jonathan Myles[2]⊕, Adam Brentnall[2], Rhian Gabe[2], Stephen Duffy[2], Sophie Williams[1], Simon Tiberi[1,3]

**1** The Royal London Hospital, Barts Health NHS Trust, Whitechapel, London, United Kingdom, **2** Wolfson Institute of Population Health, Queen Mary University of London, London, United Kingdom, **3** Blizard Institute, Barts and the London School of Medicine and Dentistry, Queen Mary University of London, London, United Kingdom

⊕ These authors contributed equally to this work.
* t.crocker-buque@nhs.net

**Data Availability Statement:** Data cannot be shared publicly because they are health services management data. Data are available from the

## Abstract

As SARS-CoV-2 infections continue to cause hospital admissions around the world, there is a continued need to accurately assess those at highest risk of death to guide resource use and clinical management. The ISARIC 4C mortality score provides mortality risk prediction at admission to hospital based on demographic and physiological parameters. Here we evaluate dynamic use of the 4C score at different points following admission. Score components were extracted for 6,373 patients admitted to Barts Health NHS Trust hospitals between 1st August 2020 and 19th July 2021 and total score calculated every 48 hours for 28 days. Area under the receiver operating characteristic (AUC) statistics were used to evaluate discrimination of the score at admission and subsequent inpatient days. Patients who were still in hospital at day 6 were more likely to die if they had a higher score at day 6 than others also still in hospital who had the same score at admission. Discrimination of dynamic scoring in those still in hospital was superior with the area under the curve 0.71 (95% CI 0.69–0.74) at admission and 0.82 (0.80–0.85) by day 8. Clinically useful changes in the dynamic parts of the score are unlikely to be associated with subject-level measurements. Dynamic use of the ISARIC 4C score is likely to provide accurate and timely information on mortality risk during a patient's hospital admission.

## Introduction

The global pandemic caused by severe acute respiratory coronavirus 2 (SARS-CoV-2) infection and resulting COVID-19 disease continues, including in the UK, where an average of around 1,000 patients per day were admitted with COVID-19 during August and September 2021, with many more seen and discharged from emergency departments [1]. The mean length of admission resulting for a COVID-19 infection in the UK National Health Service (NHS) is 8–9 days (compared to 1 day for an acute medical admission), during which time

Barts Health NHS Trust Institutional Data Access (Data Warehouse). The data for this are held within the Barts Health NHS Trust Data Warehouse and were analysed by a team at Queen Mary University of London using an existing data sharing agreement between the two institutions. The data are contained within the Trusts wider COVID-19 patient database that is used for analysis and evaluation of service delivery of hospital services. We do not have permission from either Barts Health or the Queen Mary University of London Joint Research Management Office to place the data in a public repository, however the data can be made available to researchers looking to further evaluate and validate our findings by request to the Joint Research Management Office at research. governance@qmul.ac.uk.

**Funding:** The authors received no specific funding for this work.

**Competing interests:** The authors have declared that no competing interests exist.

patients may deteriorate to require critical care, resulting in a longer admission (15–16 days), while others will improve and be discharged [2]. This combination is likely to put increasing pressure on health services through winter months particularly as population based public health measures are reduced [3].

The ISARIC 4C mortality score provides an indication of mortality risk at admission based on demographic and physiological parameters, derived from a national level population cohort study in the UK [4,5]. Components and score weights are shown in Table 1. Total scores were grouped into low risk (score 0–3, mortality rate 1.2%), intermediate risk (score 4–8, 9.9% mortality), high risk (score 9–14, 31.4% mortality), and very high risk (score ≥15, mortality 61.5%).

The score provides useful, early information enabling clinicians to make decisions around treatment requirements and safe discharge. Barts Health National Health Service Trust (BHNHST) provides health services through 5 hospital sites across east London to a demographically and socio-economically diverse population of more than 2.5 million people, and we have previously evaluated the score in our local population [6]. However, although the score provides useful information on admission, there is a need to better understand how a patient's risk evolves during a hospital stay to support health services planning. We also sought to identify whether there was a point during patients with long admissions where repeating the scoring would be clinically useful, as currently it is only validated for use at admission to hospital. Therefore, our aim was to study the dynamic use of the ISARIC 4C mortality score at

**Table 1. Demographic and clinical components of the ISARIC 4C COVID-19 mortality score and associated score weights.**

| Variable | | Score |
|---|---|---|
| Age (years) | <50 | - |
| | 50–59 | +2 |
| | 60–69 | +4 |
| | 70–79 | +6 |
| | ≥80 | +7 |
| Sex at birth | Female | - |
| | Male | +1 |
| Number of co-morbidities* | 0 | - |
| | 1 | +1 |
| | ≥2 | +2 |
| Respiratory rate (breaths per minute) | <20 | - |
| | 20–29 | +1 |
| | ≥30 | +2 |
| Peripheral oxygen saturation on room air (%) | ≥92 | - |
| | <92 | +2 |
| Glasgow coma scale score | 15 | - |
| | <15 | +2 |
| Urea (mmol/L) | <7 | - |
| | 7–14 | +1 |
| | >14 | +3 |
| C-reactive protein (mg/L) | <50 | - |
| | 50–99 | +1 |
| | ≥100 | +2 |

*As defined by the Charlson comorbidity index, with the addition of clinician defined obesity.

different points following a patient's admission with COVID-19 to evaluate whether additional information on mortality risk can be established through admission.

## Methods

Details of the construction of the dataset have previously been published [6]. In summary, a range of demographic, prognostic and clinical factors were extracted from the BHNHST Electronic Health Record System (EHR) for all patients with a laboratory confirmed reverse transcription polymerase chain reaction (RT-PCR) positive swab result for SARS-CoV-2 from any anatomical site. All patients aged 18 years or more admitted to three hospital sites from BHNHST and with a positive test recorded up to 7 days before, or 7 days after their first admission were included. The resulting cohort for this study included 6,373 patients admitted between 1st August 2020 and 19th July 2021. The primary outcome measure was death, defined as all-cause mortality within 28-days of admission and all patients were followed-up for at least 28 days.

### Statistical methods

Characteristics included in the ISARIC 4C mortality score (4C) were extracted for each included patient for the duration of their admission as described in the paper by Knight et al. [4]. We calculated a dynamic version of 4C denoted (4CD) every 48 hours for 28 days by applying the same score at different time points using the following method: for a patient alive and not in ICU at a time (t) hours after admission, we considered all measurements of the time-dependent variables in 4CD (respiratory rate, oxygen saturation (%), Glasgow Coma Scale (GCS) score, urea (mmol/L), C-Reactive Protein (mmol/L; CRP), lymphocyte count (n x 10^9/L) taken during the interval (*t-48,t hours)*. If no measurements of a given variable were taken during this interval, we found the most recent 48-hour interval during which measurements on that variable were taken and used that interval. If there was no recorded measurement of a score components, we assumed no increased risk for that component. Where multiple recordings of a variable were made, we calculated and used the mean value. These values were incorporated into the 4CD score calculator along with the non-time dependent variables (age, sex, number of comorbidities) for each patient at each time-point. Receiver operating characteristic (ROC) curves were plotted to compare sensitivity and specificity of the 4C at admission and 4CD at relevant time points. AUC statistics were based on these with 95% DeLong confidence interval. Furthermore, in all patients still alive, in hospital, and not in ICU at day 8 we analysed the extent to which modifiable parameters (respiratory rate, oxygen saturation, GCS, urea and CRP) contributed to the change in score. The score change was calculated by the difference in 4CD between day 8 and entry alongside the mean change in 5 time-dependent variables (respiratory rate, urea, oxygen saturations, GCS and CRP). 95% CIs for mean changes in these were estimated a non-parametric bootstrap (5000 resamples). The distribution of 4CD factors at admission and day 8 was tabulated based on risk groups at day 8. A sensitivity analysis was used to do the same analysis only including those with complete data for all components at both time points. Cutpoints for continuous variables were based on the 4C definitions. Cutpoints based on 4C scores were based on recommendations from ISARIC. We have presented examples of score change over time for the individual score with the most number of patients within the intermediate, high and very high score groups.

Linear regression and Wald tests were used to investigate the association between age, sex and any comorbidities on change from baseline to day 8 (in those still in hospital) in inflammation components of 4CD (score due to GCS, Urea, and CRP), or respiratory components

(score due to oxygen saturation and respiratory rate). All statistical analyses were undertaken in R version 4.1.1.

### Ethical approval

The study proposal was submitted the Joint Research Management Office at Queen Mary, University of London, who reviewed the methods and determined that as it is an evaluation of routinely collected hospital data it did not require ethics committee review and approval. The study was registered as a health services evaluation with the Barts Health NHS Trust Clinical Effectiveness Unit (reference 11121). Individual patient consent was not required or sought.

### Results

Baseline characteristics of the 6,373 included patients are shown in Table 2. Unlike many cohorts of patients in hospital with SARS-CoV2 infection the cohort had good representation from all major ethnic groups.

To explore the change in score between survivors and decedents we created graphics for score changes during admission for each total score value. Here we present figures for the the score with the largest number of patients in the intermediate (score = 5), high (score = 10) and very high (score = 15) score groups to illustrate how the underlying score total changes during admission in patients with different risk profiles. **Fig 1** considers the distribution of 4CD in patients with a score of 10 (n = 1,937) at admission and still in hospital over time, by subsequent mortality status. Equivalent figures for scores of 5 (n = 1,079) and 15 (n = 438) are presented in the (S1 and S2 Figs). These demonstrate that it is possible to better assess prognosis in groups of patients with identical scores at entry, by dynamically updating 4CD.

S1 Table gives statistics on baseline characteristics of these sub-cohorts. The main factors driving the higher scores were older age, being male, having a greater number of comorbidities and being more likely to be confused (GCS<15). The highest baseline score group showed much greater concentrations of urea and were more likely to have high CRP levels. However, there was very little variation in respiratory rate or oxygen saturation between the groups.

We undertook further analysis in patients at still in hospital 8 days after admission. **Fig 2** shows distinct patterns in survival for patients classified at day 8 using 4CD into low (0–3), intermediate (4–8), high (9–14) and very high (15+) risk groups, with high mortality in the very high-risk group, and moderate mortality over a longer period in the high group. The ROC obtained by classifying patients using rules based on the 4CD at day 8 and the 4C score at admission is shown in the (S3 Fig). The AUC obtained with the 4CD is 0.823 (95% CI 0.800–0.845) and that with the 4C score at admission is 0.713 (95% CI (0.686–0.740), S3 Fig is the equivalent plot with 4CD calculated at day 16, again showing a much higher AUC than using the admission score alone.

This pattern of superior risk assessment using up to date information and dynamic application of 4CD was also observed at all time points after admission (Table 3).

In further analysis to identify which time-dependent components contributed most to the change in score changes in Urea and CRP showed evidence to have been larger contributors to change, with a lesser contribution from respiratory rate (Table 4).

A comparison of the distribution of 4C components at day 8 and entry by score band at day 8 is shown in Table 5. Those in the highest risk group at day 8 were predominantly aged 80 years or more, and male. As a group they had seen a worsening in their prognosis associated with Urea and CRP, but less change in respiratory rate, and a generally improved profile for oxygen saturation (which is likely partly due to receiving oxygen in hospital). In contrast, the

**Table 2. A table showing the demographic and clinical characteristics of all included patients at admission, and divided by mortality status at 28 days, including all components of the ISARIC 4C score as well as ethnic group.**

| Variable | Category | Alive at 28 days n = 5,027 (%) | Dead within 28 days n = 1,346 (%) | Total n = 6,373 (%) |
|---|---|---|---|---|
| Sex | Female | 2162 (43) | 519 (38.6) | 2681 (42.1) |
| | Male | 2865 (57) | 827 (61.4) | 3692 (57.9) |
| Age | Under 50 | 1546 (30.8) | 31 (2.3) | 1577 (24.7) |
| | 50–59 | 996 (19.8) | 84 (6.2) | 1080 (16.9) |
| | 60–69 | 1023 (20.4) | 250 (18.6) | 1273 (20) |
| | 70–79 | 753 (15) | 322 (23.9) | 1075 (16.9) |
| | 80 and over | 709 (14.1) | 659 (49) | 1368 (21.5) |
| Ethnic group | White | 1563 (31.1) | 518 (38.5) | 2081 (32.7) |
| | Asian (Indian, Bangladeshi, Pakistani) | 1422 (28.3) | 387 (28.8) | 1809 (28.4) |
| | Asian (other) | 372 (7.4) | 73 (5.4) | 445 (7) |
| | Black | 671 (13.3) | 178 (13.2) | 849 (13.3) |
| | Other | 391 (7.8) | 67 (5) | 458 (7.2) |
| | Mixed | 40 (0.8) | 8 (0.6) | 48 (0.8) |
| | Unknown | 568 (11.3) | 115 (8.5) | 683 (10.7) |
| Co-morbidities* | 0 | 2968 (59.0) | 382 (28.4) | 3350 (52.6) |
| | 1 | 738 (14.7) | 239 (17.8) | 977 (15.3) |
| | 2 or more | 1321 (26.3) | 725 (53.9) | 2046 (32.1) |
| Respiratory rate (breaths per minute) | 0 to 19 | 1826 (36.3) | 314 (23.3) | 2140 (33.6) |
| | 20 to 29 | 2441 (48.6) | 707 (52.5) | 3148 (49.4) |
| | 30 or more | 476 (9.5) | 229 (17) | 705 (11.1) |
| | Unknown | 284 (5.6) | 96 (7.1) | 380 (6) |
| Oxygen saturation (%) | Less than 92 | 2783 (55.4) | 503 (37.4) | 3286 (51.6) |
| | 92 or more | 477 (9.5) | 145 (10.8) | 622 (9.8) |
| | Unknown | 1767 (35.2) | 698 (51.9) | 2465 (38.7) |
| Glasgow Coma Scale (score out of 15) | 15 | 4338 (86.3) | 895 (66.5) | 5233 (82.1) |
| | Less than 15 | 394 (7.8) | 358 (26.6) | 752 (11.8) |
| | Unknown | 295 (5.9) | 93 (6.9) | 388 (6.1) |
| Urea (mmol/L) | Less than 7 | 3109 (61.8) | 416 (30.9) | 3525 (55.3) |
| | 7 to 14 | 790 (15.7) | 426 (31.6) | 1216 (19.1) |
| | Greater than 14 | 368 (7.3) | 302 (22.4) | 670 (10.5) |
| | Unknown | 760 (15.1) | 202 (15) | 962 (15.1) |
| C-reactive protein (mmol/L) | less than 50 | 1277 (25.4) | 216 (16) | 1493 (23.4) |
| | 50–99 | 1061 (21.1) | 270 (20.1) | 1331 (20.9) |
| | 100 or greater | 1475 (29.3) | 566 (42.1) | 2041 (32) |
| | Unknown | 1214 (24.1) | 294 (21.8) | 1508 (23.7) |

*As defined by the Charlson Co-morbidities index with the addition of clinician defined obesity.

lower risk groups had all a better profile at day 8 in most of the 4CD components, except oxygen saturation. Findings were unchanged in a complete case analysis (S2 Table).

Finally, we did not find any evidence that clinically useful changes in the dynamic parts of the score were associated with subject-level measurements. Specifically, age, sex and number of co-morbidities were not associated with changes to CRP, urea or GCS. We observed a slight association of age with respiratory score change (p<0.001), however, this was too small to be of clinical utility (increase in age by 10 years associated with a respiratory score change of approximately 0.09).

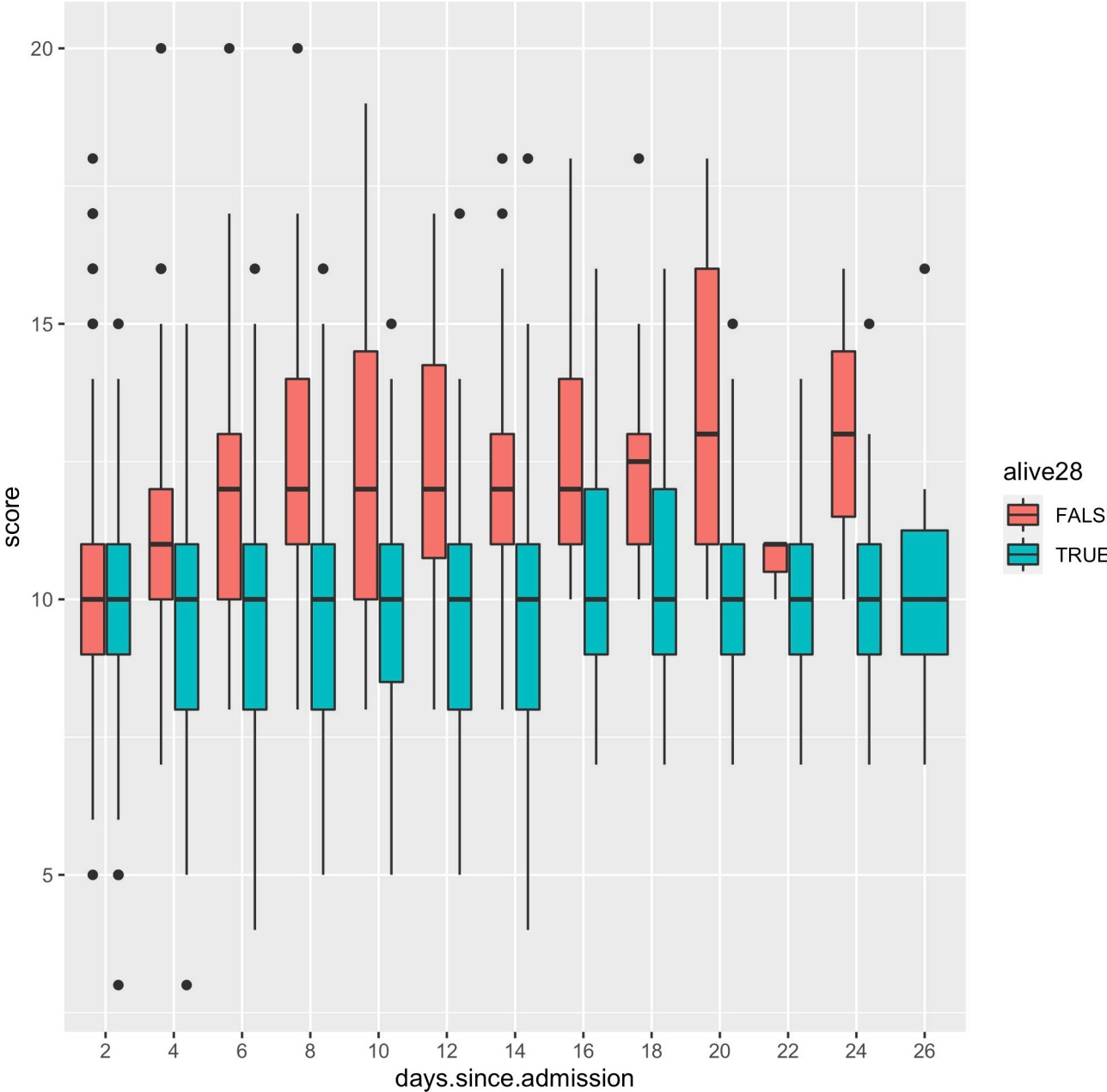

**Fig 1. Boxplot distributions of 4CD in 48-hour intervals amongst survivors (turquoise) and decedents (red) in those still in hospital and who had a score of 10 at admission (high risk).**

## Discussion

Due to the continuing pressure COVID-19 is putting on health services there remains a need for accurate clinical risk scoring to enable appropriate use of resources, as well as to identify and intervene early in deteriorating patients [7,8]. The ISARIC 4C score has been shown to provide accurate mortality risk predictions in COVID-19 patients at admission, but here we demonstrate that repeated, dynamic application of the score increases accurate mortality prediction as risk changes during hospital admission. Repeating the ISARIC 4C score at any

## survival in days after 8 days in hospital by ICD band

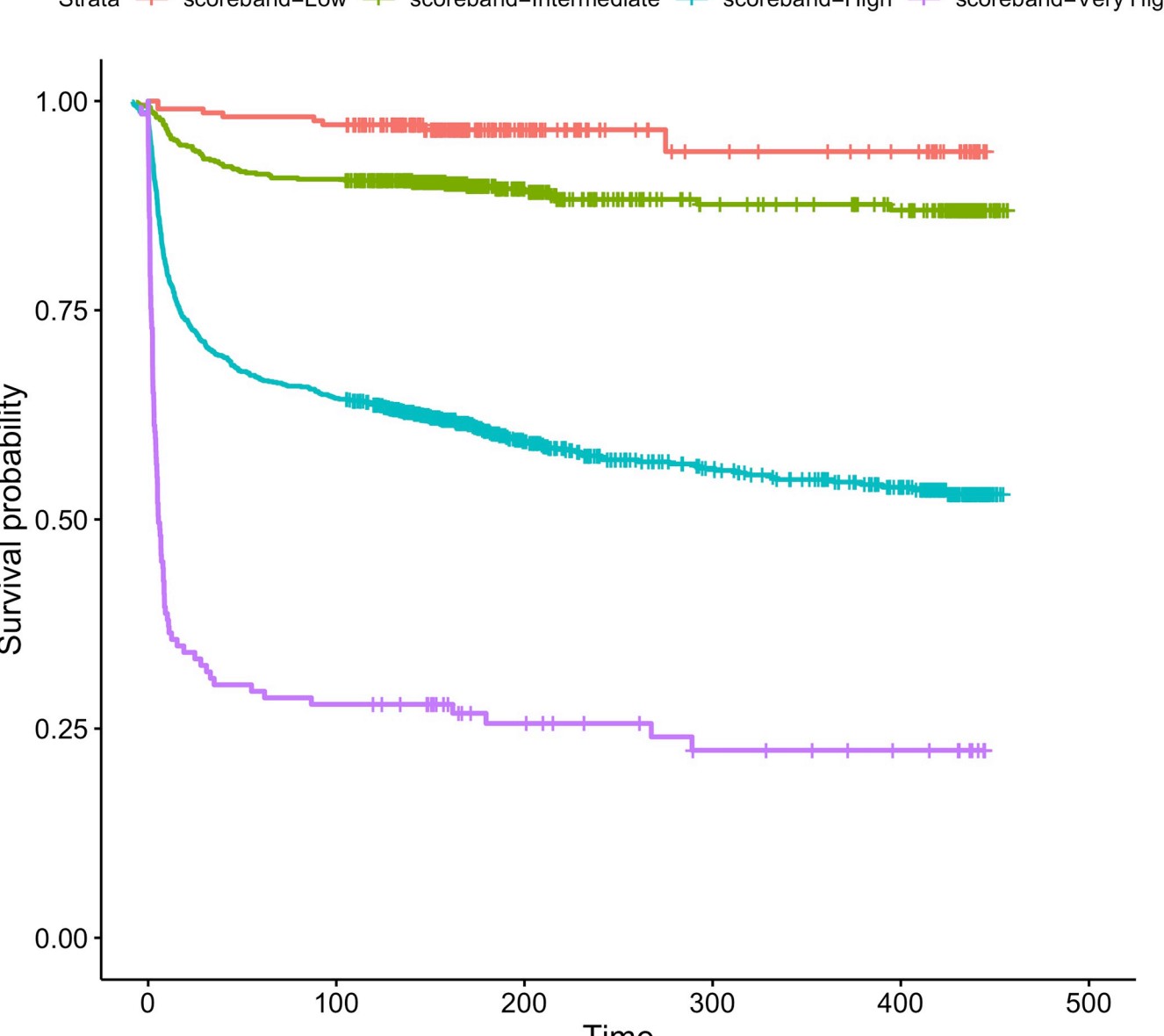

**Fig 2. Survival in patients grouped into low (0–3), intermediate (4–8), high (9–14) and very high (15+) 4CD score groups in patients alive and not in ICU at 8 days.**

further day after admission increases the sensitivity and specificity of the score in identifying those with the highest risk of dying. This supports previous evidence that dynamic scoring during hospital admission involving biochemical parameters can increase accuracy in predicting mortality [9]. Repeated scores may be exploited to change management of the individual patient in response to change or failure to change in the score. However, the area under the curve using dynamic scoring remains 0.8, which is adequate for a test of this kind, but leaves residual risk of false positive or false negative categorisation. The risk of falsely scoring a

**Table 3. Area under the curve of the receiver operating characteristic using 4CD score calculated every 2 days compared to 4C score on admission in all patients alive and not in ICU, with 95% Confidence Interval.**

| Day post admission | 4CD score AUC* (95% CI) | Admission 4C score AUC* ((95% CI) |
|---|---|---|
| 2 | 0.83 (0.82 to 0.84) | 0.81 (0.80 to 0.83) |
| 4 | 0.85 (0.83 to 0.86) | 0.79 (0.77 to 0.80) |
| 6 | 0.84 (0.82 to 0.86) | 0.76 (0.74 to 0.78) |
| 8 | 0.82 (0.80 to 0.85) | 0.71 (0.69 to 0.74) |
| 10 | 0.81 (0.78 to 0.83) | 0.68 (0.64 to 0.71) |
| 12 | 0.81 (0.78 to 0.84) | 0.63 (0.59 to 0.67) |
| 14 | 0.83 (0.79 to 0.86) | 0.60 (0.55 to 0.65) |
| 16 | 0.82 (0.77 to 0.86) | 0.59 (0.53 to 0.65) |
| 18 | 0.80 (0.74 to 0.86) | 0.55 (0.47 to 0.63) |
| 20 | 0.84 (0.78 to 0.91) | 0.53 (0.44 to 0.62) |
| 22 | 0.80 (0.73 to 0.88) | 0.60 (0.47 to 0.74) |

* p value comparing 4CD with admission score <0.001 at all time points.

patient as being lower risk could result in false reassurance. Therefore, this score should only be used as one tool within a full clinical risk assessment of an admitted patient.

We have demonstrated that the change in score is being driven by increasing CRP and urea, with associated worsening respiratory and neurological function. This supports the wider evidence that increasing systemic inflammatory response is associated with worse outcomes for patients [10]. Urea also forms a component of the CURB-65 clinical risk core for community acquired pneumonia and is known to be associated with worsening fluid status and renal dysfunction [11]. Existing inpatient systems to monitor patients for deterioration, such as the National Early Warning Score 2 (NEWS2) only include vital sign measurements, which cover oxygen saturation, respiratory rate, and GCS, along with oxygen use, pulse and blood pressure [12]. However, NEWS2 doesn't include any biochemical measurements, nor age or sex, which are known to be significant contributors to COVID-19 mortality [13]. Other studies have added age to NEWS parameters to improve accuracy, but these are lower than the dynamic application of ISARIC 4C described here [14].

These results have been derived from a large, ethnically, and socio-economically diverse population of hospital in-patients, with a wide range of underlying co-morbidities. The window for included patients covers the introduction of core therapeutic interventions, such the use of the anti-viral remdesivir, systemic steroids (dexamethasone), interleukin-6 receptor blockade (tocilizumab/sarilumab), REGEN-COV (casirivimab and imdevimab) amongst others as identified through trials such as the RECOVERY trial [15]. It also includes patients infected with both alpha and delta SARS-CoV-2 variant viruses [16]. It includes a cohort of patients prior to widespread vaccination being introduced in the UK (from December 2020), those partially and completely vaccinated between December and July 2021 [17]. Vaccinated patients are less likely to suffer from severe disease, therefore, it is likely that ISARIC 4C will remain of most relevance in patients who are unvaccinated and more likely to have high levels of viral replication and subsequent inflammatory response [18]. The ISARIC 4C score already features as a recommended tool for risk assessment when prescribing Remdesivir, 4C score ≥4 (those with a lower score 0–3 being likely to recover without treatment), however there is potential future scope for the ISARIC 4C score to assist in guiding therapeutic decisions for other treatments [19].

**Table 4. Mean change (95%CI) in 4C components associated with an overall 4CD change between entry and day 8, in patients still in hospital at day 8.**

| Score change | Number of patients | Respiratory rate score mean change | Urea score mean change | Oxygen saturation score mean change | GCS score mean change | CRP score mean change |
|---|---|---|---|---|---|---|
| | | Range: 0–2 | Range: 0–3 | Range: 0–2 | Range: 0–2 | Range: 0–2 |
| -5 | 34 | -0.65 (-0.91 to -0.44) | -1.53 (-1.94 to -1.15) | -0.88 (-1.31 to -0.62) | -0.47 (-0.88 to -0.29) | -1.52 (-1.76 to -1.21) |
| -4 | 68 | -0.69 (-0.87 to -0.56) | -0.88 (-1.16 to -0.64) | -0.83 (-1.11 to -0.62) | -0.44 (-0.70 to -0.29) | -1.18 (-1.37 to -0.99) |
| -3 | 134 | -0.78 (-0.88 to -0.69) | -0.36 (-0.54 to -0.20) | -0.35 (-0.52 to -0.22) | -0.27 (-0.43 to -0.15) | -1.18 (-1.32 to -1.04) |
| -2 | 224 | -0.61 (-0.70 to -0.54) | 0.03 (-0.09 to 0.14) | -0.20 (-0.33 to -0.11) | -0.21 (-0.31 to -0.13) | -1.04 (-1.16 to -0.92) |
| -1 | 287 | -0.53 (-0.61 to -0.47) | 0.07 (-0.03 to 0.15) | 0.04 (-0.04 to 0.12) | -0.08 (-0.17 to -0.02) | -0.49 (-0.59 to -0.39) |
| 0 | 338 | -0.23 (-0.29 to -0.18) | 0.26 (0.17 to 0.34) | 0.15 (0.08 to 0.23) | -0.02 (-0.07 to 0.02) | -0.15 (-0.25 to -0.07) |
| 1 | 240 | -0.01 (-0.10 to 0.06) | 0.43 (0.34 to 0.53) | 0.36 (0.25 to 0.46) | 0.03 (-0.05 to 0.10) | 0.23 (0.09 to 0.34) |
| 2 | 171 | -0.09 (-0.20 to -0.01) | 0.71 (0.58 to 0.84) | 0.57 (0.43 to 0.71) | 0.29 (0.16 to 0.41) | 0.55 (0.39 to 0.68) |
| 3 | 82 | 0.17 (0.01 to 0.29) | 1.00 (0.77 to 1.24) | 0.73 (0.49 to 0.94) | 0.44 (0.24 to 0.61) | 0.72 (0.51 to 0.89) |
| 4 | 34 | 0.30 (0.06 to 0.48) | 1.36 (0.94 to 1.73) | 0.97 (0.58 to 1.23) | 0.61 (0.18 to 0.91) | 0.94 (0.64 to 1.15) |

**Table 5. Summary statistics on score components by risk group at day 8 showing values at baseline on admission and at day 8.** Risk groups: Low (0–3), intermediate (4–8), high (9–14) and very high (15+).

| Variable | Category | Low n = 184 (%) | | Medium n = 562 (%) | | High n = 835 (%) | | Very High n = 110 (%) | |
|---|---|---|---|---|---|---|---|---|---|
| | | Admission | Day 8 | Admission | Day 8 | Admission | Day 8 | Admission | Day 8 |
| Sex | Female | 88 (47.8) | | 258 (45.9) | | 362 (43.4) | | 26 (23.6) | |
| | Male | 96 (52.2) | | 304 (54.1) | | 473 (56.6) | | 84 (76.4) | |
| Age | Under 50 | 148 (80.4) | | 42 (7.5) | | 2 (0.2) | | 0 (0.0) | |
| | 50–59 | 36 (19.6) | | 153 (27.2) | | 8 (1.0) | | 0 (0.0) | |
| | 60–69 | 0 (0.0) | | 219 (39.0) | | 94 (11.3) | | 0 (0.0) | |
| | 70–79 | 0 (0.0) | | 98 (17.4) | | 268 (32.1) | | 11 (10.0) | |
| | 80 and over | 0 (0.0) | | 50 (8.9) | | 463 (55.4) | | 99 (90.0) | |
| Co-morbidities | 0 | 158 (85.9) | | 347 (61.7) | | 191 (22.9) | | 8 (7.3) | |
| | 1 | 14 (7.6) | | 96 (17.1) | | 160 (19.2) | | 12 (10.9) | |
| | 2 or more | 12 (6.5) | | 119 (21.2) | | 484 (58.0) | | 90 (81.8) | |
| Respiratory rate (breaths per minute) | 0 to 19 | 58 (34.1) | 145 (80.1) | 182 (35.3) | 402 (71.9) | 300 (39.4) | 582 (69.9) | 43 (43.4) | 42 (38.2) |
| | 20 to 29 | 95 (55.9) | 36 (19.9) | 289 (56.1) | 148 (26.5) | 417 (54.7) | 237 (28.5) | 50 (50.5) | 67 (60.9) |
| | 30 or more | 17 (10.0) | 0 (0.0) | 44 (8.5) | 9 (1.6) | 45 (5.9) | 14 (1.7) | 6 (6.1) | 1 (0.9) |
| | Unknown | 14 (7.6) | 3 (1.6) | 47 (8.4) | 3 (0.5) | 73 (8.7) | 2 (0.2) | 11 (10) | 0 (0) |
| Oxygen saturation (%) | Less than 92 | 93 (80.9) | 164 (97.6) | 247 (78.7) | 472 (90.9) | 447 (84.7) | 636 (81.3) | 61 (88.4) | 48 (43.6) |
| | 92 or more | 22 (19.1) | 4 (2.4) | 67 (21.3) | 47 (9.1) | 81 (15.3) | 146 (18.7) | 8 (11.6) | 62 (56.4) |
| | Unknown | 69 (37.5) | 16 (8.7) | 248 (44.1) | 43 (7.7) | 307 (36.8) | 53 (6.3) | 41 (37.3) | 0 (0) |
| Glasgow Coma Scale (score out of 15) | 15 | 163 (95.9) | 178 (98.9) | 471 (91.5) | 533 (95.2) | 562 (73.9) | 637 (76.5) | 53 (53.5) | 39 (35.5) |
| | Less than 15 | 7 (4.1) | 2 (1.1) | 44 (8.5) | 27 (4.8) | 198 (26.1) | 196 (23.5) | 46 (46.5) | 71 (64.5) |
| | Unknown | 14 (7.6) | 4 (2.2) | 47 (8.4) | 2 (0.4) | 75 (9) | 2 (0.2) | 11 (10) | 0 (0) |
| Urea (mmol/L) | Less than 7 | 143 (92.9) | 134 (75.3) | 329 (72.0) | 317 (57.5) | 283 (40.0) | 232 (28.3) | 13 (15.1) | 4 (3.7) |
| | 7 to 14 | 10 (6.5) | 44 (24.7) | 90 (19.7) | 219 (39.7) | 279 (39.4) | 444 (54.1) | 38 (44.2) | 37 (33.9) |
| | Greater than 14 | 1 (0.6) | 0 (0.0) | 38 (8.3) | 15 (2.7) | 146 (20.6) | 145 (17.7) | 35 (40.7) | 68 (62.4) |
| | Unknown | 30 (16.3) | 6 (3.3) | 105 (18.7) | 11 (2) | 127 (15.2) | 14 (1.7) | 24 (21.8) | 1 (0.9) |
| C-reactive protein (mmol/L) | less than 50 | 53 (39.0) | 152 (86.9) | 126 (30.1) | 384 (71.8) | 201 (31.1) | 418 (51.1) | 22 (27.2) | 19 (17.4) |
| | 50–99 | 31 (22.8) | 17 (9.7) | 120 (28.6) | 94 (17.6) | 193 (29.8) | 237 (29.0) | 27 (33.3) | 28 (25.7) |
| | 100 or greater | 52 (38.2) | 6 (3.4) | 173 (41.3) | 57 (10.7) | 253 (39.1) | 163 (19.9) | 32 (39.5) | 62 (56.9) |
| | Unknown | 48 (26.1) | 9 (4.9) | 143 (25.4) | 27 (4.8) | 188 (22.5) | 17 (2) | 29 (26.4) | 1 (0.9) |

### Limitations

Limitations include lack of inclusion of prescribing and therapeutic data within the dataset, which cannot be factored into the appropriateness of using the score. We only included parameters within the ISARIC 4C model and did not consider other factors which may contribute to risk. Included patients were from a defined geographic area and findings may not be generalisable to other contexts. We did not have information on the vaccine status of patients included in this study.

### Conclusion

Dynamic use of the ISARIC 4C score can provide accurate and timely information on mortality risk during a patient's hospital admission.

## Supporting information

**S1 Fig. Showing mean and range of ISARIC 4C scores at 48-hour intervals amongst survivors (turquoise) and decedents (red) with a score of 5 at admission.**
(DOCX)

**S2 Fig. Showing mean and range of ISARIC 4C scores at 48-hour intervals amongst survivors (turquoise) and decedents (red) with a score of 15 at admission.**
(DOCX)

**S3 Fig. Receiver operating characteristic curves comparing sensitivity and specificity of mortality risk using 4C score at admission (green) and at day 16 (blue).**
(DOCX)

**S1 Table. Baseline characteristics on score components at admission in patient grouped by admission 4C scores of 5, 10 and 15.**
(DOCX)

**S2 Table. 4CD components (n, %) in patients with low (0–3), intermediate (4–8), high (9–14) and very high (15+) scores at day 8, for those with complete data at both time points in all measurement.**
(DOCX)

## Author Contributions

**Conceptualization:** Tim Crocker-Buque, Adam Brentnall, Rhian Gabe, Stephen Duffy, Sophie Williams, Simon Tiberi.

**Data curation:** Tim Crocker-Buque, Jonathan Myles, Adam Brentnall, Rhian Gabe, Sophie Williams.

**Formal analysis:** Tim Crocker-Buque, Jonathan Myles, Adam Brentnall, Stephen Duffy, Sophie Williams, Simon Tiberi.

**Investigation:** Simon Tiberi.

**Methodology:** Tim Crocker-Buque, Jonathan Myles, Adam Brentnall, Rhian Gabe, Stephen Duffy, Sophie Williams, Simon Tiberi.

**Project administration:** Tim Crocker-Buque.

**Software:** Jonathan Myles.

**Supervision:** Rhian Gabe, Stephen Duffy, Simon Tiberi.

**Writing – original draft:** Tim Crocker-Buque, Adam Brentnall.

**Writing – review & editing:** Tim Crocker-Buque, Jonathan Myles, Adam Brentnall, Rhian Gabe, Stephen Duffy, Sophie Williams, Simon Tiberi.

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
