## [Decision Letter · Decision Letter 0]

8 Jun 2022

PONE-D-22-13262Using ISARIC 4C mortality score to predict dynamic changes in mortality risk in COVID-19 patients during hospital admission.PLOS ONE

Dear Dr. Crocker-Buque,

Thank you for submitting your manuscript to PLOS ONE. After careful consideration, we feel that it has merit but does not fully meet PLOS ONE’s publication criteria as it currently stands. Therefore, we invite you to submit a revised version of the manuscript that addresses the points raised during the review process.

We look forward to receiving your revised manuscript.

Kind regards,

Jordi Camps

Academic Editor

PLOS ONE

Journal Requirements:

Additional Editor Comments:

It is an interesting article that provides useful information for the management of patients with COVID-19. However, Reviewer #1 is correct that the presentation is very confusing. The numbering of the supplementary figures does not correspond to the text. Normally, this formal sloppiness would lead me to decide to reject the article, but given the importance of having as many data as possible on this disease, I decide to give the authors another chance and recommend that they submit a new revised version that answers ALL the questions from the reviewers. On the other hand, AUC's of 0.80 are good, but not great. There will be a percentage of falsely positive or negative cases. How do the authors propose that this problem be addressed?

Reviewers' comments:

Reviewer's Responses to Questions

**Comments to the Author**

1. Is the manuscript technically sound, and do the data support the conclusions?

Reviewer #1: Yes

Reviewer #2: Yes

2. Has the statistical analysis been performed appropriately and rigorously? 

Reviewer #1: I Don't Know

Reviewer #2: Yes

3. Have the authors made all data underlying the findings in their manuscript fully available?

Reviewer #1: Yes

Reviewer #2: Yes

4. Is the manuscript presented in an intelligible fashion and written in standard English?

Reviewer #1: Yes

Reviewer #2: Yes

5. Review Comments to the Author

Reviewer #1: Thank you for the opportunity to review this article. The authors examine the prognostic ability of ISARIC 4C by capturing it dynamically, showing that the AUC at Day 8 is 0.82, which is better than the AUC at admission of 0.71.

While I find the content of the paper to be significant, I have a few concerns about the content.

First, regarding the overall structure of the paper, there are several figures and tables that are not mentioned in the main text, making it very difficult for the reader to understand the meaning of the figures and tables and what the authors are trying to show by presenting them.

The author also asks the reader to have knowledge of events that cannot be said to be generalized, such as the components of the score, and I feel this also makes it difficult to convey the content to the reader.

Below we point out some specific issues.

1. It is obvious that the closer the point of prediction is to the time of outcome occurrence, the better the predictive ability. So, I think that dynamically evaluating this score is inherently of less clinical significance.

2. The authors hope that dynamic evaluation of this score will help in the proper allocation of medical resources and treatment management. What specifically do the authors expect that evaluating day8 scores will contribute to the proper allocation of resources? Does a good score mean less human resources? Conversely, if the score is poor, does it mean less human resources to prioritize patients who can be saved? Will you change the units you manage? It seems to me that the day8 score is unlikely to contribute to a change in strategy for resource allocation as much as checking the score at the time of admission and planning accordingly. I partially agree that Day 8 scores do contribute to treatment strategies, but there is no specific indication of which scores are associated with which pathologies and which treatments are effective.

3. Although Table 1 indirectly mentions the ISARIC4C components and cutoffs, the authors should include the scores and specifically specify their contents in the text, or create a separate Table or Figure to present them.

4. The authors state that "we assumed no increased risk for that component" for the missing values, does this mean LOCF using the previous observation point? Does this mean that the score is underestimated for patients who deteriorate? Please specify how much of the missing data actually occurred for each component.

5. In Figure 1, you describe the transition during the course of a patient with a score of 10 on admission, and in Figure 5 you present a score of 5 and in Figure 6 a score of 15 in a similar manner, but Method does not mention such a presentation beforehand, which seems very abrupt. It seems very abrupt. Also, why did the authors employ scores 5, 10, and 15?

6. Figure 2 and 3 are similarly abrupt and not mentioned in the text. Figure 3 has no more information than Table 2. Why did the authors present Figure 3?

Reviewer #2: A very interesting study, it would be interesting to see if this score could indicate the risk of mortality in other environments, countries as well.

It would be necessary to adapt the heading of Table 1. It is not clear which comorbidities have been studied, could these data be unknown in some patients?

How many of the patients studied had received vaccines against SARS CoV2, which vaccines, how many days before presenting COVID-19 symptoms, and what relationship is observed with mortality in these cases?

What is the mortality of patients who have needed ICU and what of patients who have only been admitted to the ward?

6. PLOS authors have the option to publish the peer review history of their article (what does this mean?). If published, this will include your full peer review and any attached files.

Reviewer #1: **Yes: **Hiroki Nishiwaki

Reviewer #2: No

---

## [Author Response · Author response to Decision Letter 0]

4 Jul 2022

Reviewer #1: 

Thank you for the opportunity to review this article. The authors examine the prognostic ability of ISARIC 4C by capturing it dynamically, showing that the AUC at Day 8 is 0.82, which is better than the AUC at admission of 0.71. While I find the content of the paper to be significant, I have a few concerns about the content.

First, regarding the overall structure of the paper, there are several figures and tables that are not mentioned in the main text, making it very difficult for the reader to understand the meaning of the figures and tables and what the authors are trying to show by presenting them.

We have clarified this in the text, in addition to the Editor’s comments. 

The author also asks the reader to have knowledge of events that cannot be said to be generalized, such as the components of the score, and I feel this also makes it difficult to convey the content to the reader.

We have added the score components and weighting to the introduction (Table 1) 

Below we point out some specific issues.

1. It is obvious that the closer the point of prediction is to the time of outcome occurrence, the better the predictive ability. So, I think that dynamically evaluating this score is inherently of less clinical significance.

While this is to be expected, we felt it was also useful to conclusively prove this. The clinical relevance of this work is that the ISARIC 4C score is currently only validated for use at admission. In our hospital, we proposed using it at a second point during a patient’s admission, particularly in patients who we expected to have a long admission (> 1 week), but this was rejected as lacking evidence as to when re-scoring would be appropriate and if this would provide additional information. We have added some further clarification on this in the introduction. 

2. The authors hope that dynamic evaluation of this score will help in the proper allocation of medical resources and treatment management. What specifically do the authors expect that evaluating day8 scores will contribute to the proper allocation of resources? Does a good score mean less human resources? Conversely, if the score is poor, does it mean less human resources to prioritize patients who can be saved? Will you change the units you manage? It seems to me that the day8 score is unlikely to contribute to a change in strategy for resource allocation as much as checking the score at the time of admission and planning accordingly. I partially agree that Day 8 scores do contribute to treatment strategies, but there is no specific indication of which scores are associated with which pathologies and which treatments are effective.

In the context of our hospital, we had an extremely high volume of patients, often with many hundreds of inpatients with COVID-19 at the same time. Our staff were stretched to capacity in providing care. Of those not admitted to critical care, patients receiving oxygen were admitted to a high-acuity medical admissions ward, and those on non-invasive ventilation support to the respiratory ward. However, that left us with a very large number of (mainly elderly) patients who had very long hospital stays (1-3 weeks), some of whom deteriorated and died during their admission without necessarily requiring oxygen support. Many of these patients were managed on a low-acuity elderly care ward, which had a lower staff to patient ratio than the acute medical units, as usually these patients have lower acuity admissions. Our hypothesis for the use of this score was that patients at day 8 whose scores remained high or had increased since admission could have remained in a higher acuity acute medical ward. We have not added anything into the manuscript in this regard, as our experience may not match those of other hospitals, and it would be for other clinical services to decide if the additional information provided is useful in their clinical context. 

3. Although Table 1 indirectly mentions the ISARIC4C components and cutoffs, the authors should include the scores and specifically specify their contents in the text, or create a separate Table or Figure to present them.

This has been added to the introduction as the new Table 1. 

4. The authors state that "we assumed no increased risk for that component" for the missing values, does this mean LOCF using the previous observation point? Does this mean that the score is underestimated for patients who deteriorate? Please specify how much of the missing data actually occurred for each component.

Information on missing data at entry is provided in Table 2 (i.e. total number of patients in the sample minus the total in each category). We have not presented this information separately, but it can be calculated if required as the overall amount of missing information was very low. The proportion unknown was higher in those who died within 28d for oxygen saturation, slightly higher between those who died within 28d and not for respiratory rate, GCS, similar for Urea, and slightly lower for C-reactive protein. Overall, this suggests that our approach of assuming no increased risk when the value was missing was conservative at baseline, and with full data the score is likely to have been more strongly associated with the outcome. We were therefore mostly concerned with a possible risk of bias arising in evaluation of improvement relative to baseline risk assessment due to increased completeness of the unknown variables at baseline. To help assess this a complete case analysis was done for a version of Table 6, but only using those with all available components of the score (supplementary Table S8). The results from this analysis were very similar to the full sample (Table 6).

On the other hand as the reviewer notes, there is also a risk of bias in the other direction (risk model performance is underestimated) arising from LOCF. Generally, risk components in the cohort went down through time, so using LOCF may overstate risk on average at later points, which would make the model less informative. However, the model did show improvement using later values.

5. In Figure 1, you describe the transition during the course of a patient with a score of 10 on admission, and in Figure 5 you present a score of 5 and in Figure 6 a score of 15 in a similar manner, but Method does not mention such a presentation beforehand, which seems very abrupt. It seems very abrupt. Also, why did the authors employ scores 5, 10, and 15?

Agreed – this is very abrupt and is due to editing, so apologies for that. We created figures for all scores and the underlying analysis includes all the patients with all possible scores. The Figures are included as visual examples to demonstrate the differences in behaviour of the score in intermediate, high and very high-risk scoring patients. Distribution of patients across total score values was not even due to high association between different components of the score. To provide a visual representation of the results we selected one score from each of the low, medium and high-risk groups to show differences between these group. We selected 5, 10 and 15 as the scores with the largest number of patients. We have clarified this in the text. 

6. Figure 2 and 3 are similarly abrupt and not mentioned in the text. 

Figure 2 and 3 were referenced in the text, in the paragraph at the end of page 5, but we agree this was unclear and have modified this in the updated manuscript. 

Figure 3 has no more information than Table 2. Why did the authors present Figure 3?

We felt a visual representation of these data would be helpful to the reader, particularly people with previous experience of evaluating ROC curves. However, we have moved this to the supplementary material as Figure S5. 

Reviewer #2: 

A very interesting study, it would be interesting to see if this score could indicate the risk of mortality in other environments, countries as well.

We agree and hope that the further detail about how the components of the score change during the admission will encourage it to be used by other clinical teams. 

It would be necessary to adapt the heading of Table 1. 

This has been amended.

It is not clear which comorbidities have been studied, could these data be unknown in some patients?

This is part of the Charlson Co-morbidity Index with the addition of clinician defined obesity and has been detailed in the introduction. 

How many of the patients studied had received vaccines against SARS CoV2, which vaccines, how many days before presenting COVID-19 symptoms, and what relationship is observed with mortality in these cases?

The majority of patients were included in the study prior to widespread vaccination. However, we do not know how many of the patients were vaccinated as this is not included in our data-set, and is not available within the hospital’s data warehouse as these data are held by community health services. We also wish we could have included this! This is mentioned as a limitation in the limitation section. 

What is the mortality of patients who have needed ICU and what of patients who have only been admitted to the ward?

Unfortunately, this analysis is outside the scope of this study, as we did not include data on ICU admission status in the risk analysis in this way. The ISARIC 4C score is not validated for use in patients admitted to clinical care. However, this would be worthy of further work.

---

## [Editor Report · Decision Letter 1]

24 Aug 2022

Using ISARIC 4C mortality score to predict dynamic changes in mortality risk in COVID-19 patients during hospital admission.

PONE-D-22-13262R1

Dear Dr. Crocker-Buque,

We’re pleased to inform you that your manuscript has been judged scientifically suitable for publication and will be formally accepted for publication once it meets all outstanding technical requirements.

Kind regards,

Jordi Camps

Academic Editor

PLOS ONE

Additional Editor Comments (optional):

The authors satisfactorily answered the Reviewers' comments and suggestions.
---

## [Editor Report · Acceptance letter]

8 Sep 2022

PONE-D-22-13262R1 

Using ISARIC 4C mortality score to predict dynamic changes in mortality risk in COVID-19 patients during hospital admission. 

Dear Dr. Crocker-Buque:

I'm pleased to inform you that your manuscript has been deemed suitable for publication in PLOS ONE. Congratulations! Your manuscript is now with our production department. 

Kind regards, 

on behalf of

Dr. Jordi Camps 

Academic Editor

PLOS ONE